Stochastic computing in convolutional neural network implementation: a review

Lee Yang Yang
Abdul Halim Zaini zaini@usm.my
School of Electrical and Electronic Engineering, Universiti Sains Malaysia , Nibong Tebal , Penang , Malaysia
Wang Shuihua
Electronic publication date: 2020 Nov 9
Publication date: 2020
Volume: 6
Electronic Location ID: e309
Received 2020 Aug 4; Accepted 2020 Oct 1
Copyright: ©2020 Lee and Abdul Halim
Copyright year: 2020
Copyright holder: Lee and Abdul Halim
License: This is an open access article distributed under the terms of the Creative Commons Attribution License, which permits unrestricted use, distribution, reproduction and adaptation in any medium and for any purpose provided that it is properly attributed. For attribution, the original author(s), title, publication source (PeerJ Computer Science) and either DOI or URL of the article must be cited.
License URL: https://creativecommons.org/licenses/by/4.0/

Keywords: Stochastic computing, Convolutional Neural Network, Deep learning, FPGA, IoT

Funding: School of Electrical and Electronic Engineering, Universiti Sains Malaysia 1001/PELECT/8014152 This research was funded by the School of Electrical and Electronic Engineering, Universiti Sains Malaysia (1001/PELECT/8014152). The funders had no role in study design, data collection and analysis, decision to publish, or preparation of the manuscript.

==============================
Stochastic computing (SC) is an alternative computing domain for ubiquitous deterministic computing whereby a single logic gate can perform the arithmetic operation by exploiting the nature of probability math. SC was proposed in the 1960s when binary computing was expensive. However, presently, SC started to regain interest after the widespread of deep learning application, specifically the convolutional neural network (CNN) algorithm due to its practicality in hardware implementation. Although not all computing functions can translate to the SC domain, several useful function blocks related to the CNN algorithm had been proposed and tested by researchers. An evolution of CNN, namely, binarised neural network, had also gained attention in the edge computing due to its compactness and computing efficiency. This study reviews various SC CNN hardware implementation methodologies. Firstly, we review the fundamental concepts of SC and the circuit structure and then compare the advantages and disadvantages amongst different SC methods. Finally, we conclude the overview of SC in CNN and make suggestions for widespread implementation.

Introduction

Deep learning algorithms have been widely and silently integrated into our daily life; for example, image enhancer, voice search and linguistic translation. Meanwhile, the Internet of things (IoT) has gained industrial recognition, and many applications rely on edge computing whereby data are processed on the fly rather than relayed to cloud computing for reliability and security reasons (Naveen & Kounte, 2019). People have been heavily dependent on a widely accessible central processing unit (CPU) and general-purpose graphics processing unit (GPU) for deep learning research and application deployment. Although users strive to achieve great real-time response by offloading many computationally intensive tasks, such as object recognition to edge devices, those computing devices become extremely inefficient despite the utmost priority of power efficiency in IoT. Although the field-programmable gate array (FPGA) and application-specific integrated circuit (ASIC) could overcome the power efficiency issue, economically implementing deep learning hardware logic is not ideal. Thus, researchers are trying to explore alternatives to conventional binary in this specific use case, driving the rise of stochastic computing (SC).

SC was proposed in the 1960s when the cost of implementing binary computing was prohibitive but soon ran out of favour in the semiconductor industry. Unlike binary computing, SC can perform the arithmetic operation with a single logic gate. The most evident advantage of SC is its ability to reduce the area and power draw by reducing the number of active transistors (De Aguiar & Khatri, 2015). SC is also an inherently progressive precision where the output converges from the most significant figure; thus, SC is capable of making early decision termination (EDT). Power efficiency and EDT capability make deep learning application favourable (Kim et al., 2016), particularly in convolutional neural network (CNN) application.

CNN received extensive development since its introduction in 2012 due to its unprecedented performance in object recognition. CNN model development was trending from being deep and massive (highly accurate) to responsive (fast inference). In response to the IoT requirements in edge computing, researchers had attempted to reduce the math precision to save computing resources. With a reasonable trade-off for accuracy, an extreme simplification version of CNN, that is, binarised neural network (BNN), emerged with a promising hardware implementation capability and computing efficiency, rivalling SC methodology.

SC in CNN lacks widespread attention due to its cross-disciplinary nature in the computer science study. CNN is impactful in the field of machine learning, but the rising of IoT edge computing which pursues efficient computing pushed back CNN implementation hard. While many researchers focus on innovating CNN algorithms for different use cases such as medicine and agriculture, only a few of them consider how to implement CNN realistically since CNN execution is computationally intensive by itself. Given that no comprehensive and updated review exists on this specific area, in this review paper we thus attempt to investigate and survey the SC implementation in the CNN application.

Review methodology

This review intends to answer the following research questions:

(1) What are the major developments of SC elements and SC CNN in recent years? Due to the narrow field of study in SC, the related studies are scattered, let alone the SC in CNN implementation; thus, impede the development of SC CNN without a more centralised reference, increasing difficulty in identifying the research trends.

(2) How exactly is the CNN being computed/executed in the stochastic domain? SC is a unique computing methodology which is not often being mentioned in the academic study, despite its unique advantage in the surge of CNN application. Thus, there is a need to have a big picture on the SC CNN mechanism.

(3) What are the open problems and/or opportunities to implement SC CNN? SC CNN does have its implementation hurdles. Thus, it is necessary to summarise them before moving forward in this field of research.

With the research questions in mind, we first reviewed the basic concepts of SC and CNN in modern perspectives. It is necessary to understand the background of SC and CNN due to the vastly different field of studies between them. Moreover, there is a need to aggregate the knowledge of SC elements in the face of rising trends in SC developments. Then we examined the recent developments and contributions of SC in CNN computation and compared the implementation methodologies across various recent studies. Finally, we made a conclusion and some suggestions for the future of deep learning research in the SC domain.

Search criteria

An initial search was carried out to identify an initial set of papers which have the prior works on SC and CNN in hardware implementation. The search strings were then inferred and developed as follows:

(‘Stochastic computing’) OR (‘Stochastic computing deep learning’) OR (‘Stochastic computing convolutional neural network’) OR (‘Stochastic computing neural network’) OR (‘Stochastic computing image processing’)

‘Stochastic’ alone has a lot of meaning in a wide area of study. Thus, the keyword of ‘Stochastic computing’ is a necessity to narrow down the search scope. The search strings were applied to the indexed scientific database Scopus and Web of Science (peer-reviewed). Domain-oriented databases (ACM Digital Library, IEEE Xplore Digital Library) were also referred extensively. Finally, Google Scholar (limited to peer-reviewed papers) were used to find any omitted academic literature, especially in this multi-disciplinary search scope. Peer-reviewed articles were preferred to ensure only confirmed works were to be summarised in this review paper.

Scope of review

Notably, SC is not the only methods existed for efficient CNN computing. We only cover the topics of SC and SC related to CNN computing in this review study. Many articles may not directly involve CNNs, but their novel SC elements are worthy as part of the significant SC developments and potentially useful for the future SC CNN function blocks, thus, will be mentioned in this review. Some elemental studies on CNNs were referred to understand the nature of CNN algorithms better. Some surveys on CNN implementation in FPGA merely or never discuss the SC, but they shared a similar concern on efficient CNN computation. Thus, their surveys were also considered and referred to in this review study if any.

Basic concepts

SC and CNN are different fields of studies and worth a separate explanation. Thus, SC will be described first, then secondly CNN and BNN will be explained. Lastly, the competitive relationship between SC and BNN implementations will be discussed. SC is a unique concept of computing relative to traditional binary computing and has to be understood before an in-depth discussion on SC implementation in CNN at the next section.

SC

SC is favourable in IoT application due to its extreme simplicity of computing elements, where power efficiency is of utmost priority. Unlike deterministic computing that tolerates absolutely no error, SC allows errors to a certain degree, thus the name approximate computing. SC initially decodes a binary number into a bitstream in such a way that the frequency of 1’s bit represents the magnitude of value. For example, [0,0,0,0,0,1,1,1] stochastic stream is equal to 3/8 or 0.375 because it has three 1’s bits. Then, the number can be computed in the stochastic domain with a simple logic gate instead of gate combinations in the binary domain. Finally, the stochastic stream will be converted back to binary numbers with a simple counter by counting the frequency of 1’s bit, as shown in Fig. 1.

Figure 1 Process of SC and its elements.

SC took advantage of probability math to reduce the logic components required to perform an arithmetic operation. Taking Figs. 2A and 2B as examples, in the AND gate multiplication operation, the output can be defined as: (1) S3=PS3=PS1PS2=S1×S2.

Figure 2 SC arithmetic operation.

(A)AND gate as SC unipolar multiplier. (B) MUX as SC scaled adder. (C) Uncorrelated bit streams give accurate output. (D) Correlated bit streams give inaccurate output.

In the case of addition operation, the output will be scaled by half with MUX select input with a bitstream value of 0.5. The MUX scaled adder can be defined as: (2) S4=PS3PS1+1−PS3PS2=PS1+PS22=S1+S22,PS3=0.5,

where P is the probability of the stochastic stream. The AND gate multiplier only applies to unipolar math where the real number ∈ [0,1]. In the case of bipolar math where real number ∈ [−1,1] (0’s bit decodes as −1), the XNOR gate can be used as a multiplier, whereas the same MUX can function as a bipolar adder.

The stochastic number generator (SNG) becomes the heart of the SC to perform arithmetic operations in the stochastic domain. SNG consists of a random number generator (RNG) and a comparator; both worked synchronously to generate stochastic bitstream from a given binary number. However, the RNG was the biggest challenge in the previous SC circuit design because the correlation between the operating bitstreams plays a great role in SC accuracy. An SC output will be accurate only if both working streams are not correlated. Taking Figs. 2C and 2D as examples, [0,1,1,0,1,0,0,1] and [1,1,0,0,1,0,0,1] bitstreams can represent the value of 4/8, but the output on Fig. 2D is far from accurate due to a high correlation to the opposite bitstream. The correlation index of both bitstreams can be defined as: (3) ∑i=1nS1iS2i=∑i=1nS1i×∑i=1nS2in,

where ‘S’ is the respective stochastic bitstream and ‘n’ is the bit length. Thus, the accuracy is highly dependent on the randomness and the lengths of the stochastic stream. Nevertheless, not all of the SC elements are sensitive to stochastic correlation such as MUX scaled adder (Alaghi, Qian & Hayes, 2018).

Presently, a pseudo-random RNG called a linear-feedback shift register (LFSR) is widely accepted due to its simple design and effectiveness in lowering bitstream correlation (Alaghi & Hayes, 2013). LFSR consists of a feedback XOR gate and a bit shift register as shown in Fig. 3A. The register will be initialised with a specific value, and then, the register will generate pseudo-random binary values in every bit shifting clock cycle. The binary number generated from RNG will be compared with the user input binary number. Two circuits can be used as a comparator, namely, binary comparator and weighted binary generator (WBG) as shown in Figs. 3B and 3C respectively. Both are capable of generating stochastic bitstream. After the stochastic stream passed through the stochastic logic circuits, the computed stochastic streams can be converted back to the binary domain by using a simple flip-flop counter.

Figure 3 SNG components.

(A) RNG with LFSR. (B) True comparator. (C) WBG.

SC never stops improving and keep achieving great accuracy whilst using less area and power. SNG is the major overhead of the SC circuit. As such, Ichihara et al. (2014) proposed a circular shifting technique to share LFSR. Then, (Kim, Lee & Choi, 2016a) proposed a method very similar to memoisation computing to reduce the number of LFSRs in large scale SC. Xie et al. (2017) attempted to share LFSR with wire exchange technique with additional random bit flip, whereas Joe & Kim (2019) proposed symmetrical exchange of odd wire and even wire. Even better, Salehi (2020) showed that simple wire permutation paired with WBG could deliver the lowest correlation index, thus achieving great accuracy whilst requiring fewer logic gates. Interestingly, Chen, Ting & Hayes (2018) replaced LFSR with up-counter in conjunction with WBG to take advantage of WBG binary weighting characteristics to assure SC progressive precision behaviour. As such, zero-error EDT is achievable without extra hardware cost. The WBG could also be shared partially because some WBG logics could be redundant (Yang et al., 2018).

More advanced operations, such as square, division and non-linear functions, had also gained attention and innovations to fit modern applications. The stochastic square is already in its simplest form as shown in Fig. 4A. Squaring stochastic stream can be conducted by delaying the input stream with D flip-flop before multiplying itself. In the case of a non-linear function, such as hyperbolic tangent (TanH), stochastic TanH (Stanh) uses k-state finite state machine (FSM), such as that in Fig. 4B. FSM is a class of logic circuits that will have a specific logical output pattern only if the input reached a designated sequential threshold. Stanh function can be described as: (4) StanhK,x= tanhKx2,

where K is the number of states (must be multiples of 2) and x is the input stream. Brown & Card (2001) showed that Stanh function responds closely to the true TanH function with K = 16. However, too many states will result in random walk behaviour (Kim et al., 2016); thus, an optimum amount of state for accurate reproduction of TanH function exists in the stochastic domain. An improvement in FSM design can also emulate linear and exponential functions (Najafi et al., 2017).

Figure 4 SC with advance arithmetic operations.

(A) Stochastic squaring with D flip-flop. (B) K-state FSM for Stanh function which will be widely utilised in SC CNN.

The real challenge in SC (also the missing part of SC) is the stochastic divider design. Stochastic division traditionally used FSM with extra SNG components for gradient descent approach as shown in Figs. 5A and 5B, but the gradient descent convergence time incurred inaccuracy on the output. Newer SC divider from Chen & Hayes (2016) exploited the stochastic correlation properties to perform stochastic division without using expensive SNG as shown in Figs. 5C and 5D. This event is possible because if stochastic stream p(x) and p(y) are perfectly correlated, and p(x) < p(y), then by probability math: (5) px=1,y=1=px=1.

Given that conditional probability p(x = 1|y = 1) (probability of x = 1 given that y = 1) can also be expressed as: (6) px=1|y=1=px=1,y=1py=1=pxpy,

the desired divider function on the SC domain is derived as a result. Hence, the stochastic division can be performed if both stochastic streams are perfectly correlated by evaluating the conditional probability of x and y. In the case of p(x) > p(y), the output will be clipped to a value of 1. Chu (2020) improved the circuit by using JK flip-flop, but only for unipolar SC division.

Figure 5 SC divider circuits.

(A) Former gradient descent unipolar divider. (B) Former SC bipolar divider. (C) Newer SC unipolar divider by exploiting correlation. (D) Newer SC bipolar divider by adding sign information.

The overall structure of SC is thus explained. Other than the benefit of power efficiency, SC is also inherently error resilient where accidental bit flips will not affect the overall operation of the stochastic circuits; otherwise, it could be catastrophic in deterministic computing. Secondly, SC is inherently progressive precision whereby the output value converges from the most significant figures. For instance, if the output is 0.1234, then ‘0.1…’ will show first in the stochastic compute cycles instead of ‘…4’ in the conventional binary. This characteristic is useful in specific applications, such as weather forecasting, where only the most significant figure matters in decision making. Thus, performing EDT in SC without waiting for complete computation is possible. With that said, its simplicity did come at a cost. Increasing math precision in SC also requires long bit lengths, thus increasing computing time latency by 2n folds. For instance, doubling numerical precision from 4 to 8 bits requires increasing bit length from 24 = 16 bits to 28 = 256 bits, or 24 times exponential increase in computing latency.

SC becomes unfavourable to modern computation needs due to the ever-increasing efficiency in binary computing. Nevertheless, certain niche applications can still benefit from SC topology, such as a low-density parity-check decoder in a noisy data-transmission environment; very robust image processing tasks, such as gamma correction and Sobel edge detection (Joe & Kim, 2019); and the recent interest in CNN algorithm.

CNN

With the advancement of computing technology, many applications are getting highly reliant on probabilistic computation. Deep neural network (DNN) is a widely accepted class of machine learning algorithms to process complex information, such as images and videos. The nature of DNN consists of layers of addition and multiplication of numerical weights that end up computing the overall dimensionless probability values of an output class, which in turn allows the computer to decide based on the output value. Many DNN algorithm variations exist, each for a particular purpose, such as CNN for image processing and long short-term memory for neural-linguistic processing. CNN, for example, can reduce multidimensional images into simple classes; thus, CNN is very popular in image classification and object recognition.

The most distinctive component that discriminates CNN from other DNN algorithms is its convolution layer. CNN can reduce large matrices into a single value representation, as shown in Fig. 6A, which explains its superior capability of dimensional reduction in image processing. The convolution process can be generalised as: (7) yjl=fxjl=f∑i=1nxil−1×wijl−1+bjl,

where xjl is the convolved feature of the next layer, xil−1 is the feature from the previous layer, wijl−1 is the kernel weight matrix, and bjl is bias. ‘l’ is the layer number, ‘i’ denotes scan window number, ‘n’ is the total number of scan window, and ‘j’ is the depth of next feature map. After the convolution, the activation function fxjl exists, which can be a linear or non-linear function. Rectified linear unit (ReLU) and Tanh are just the names of a few popular activation functions. The final product yjl will be aggregated, and the process repeats, depending on the structure of the CNN model.

Figure 6 CNN’s convolution and activation.

(A) Matrix convolution. (B) Neural network model after the convolution. (C) Architecture of classical LeNet-5 CNN.

The convolution and activation layers are fundamental in CNN, albeit other optional layers exist, such as normalisation layer (to reduce output variance) (Ioffe & Szegedy, 2015), pooling layer (to save memory), and dropout layer (to prevent overfitting) (Hinton et al., 2012). At the end of convolution, the convolved pixel matrix will be flattened into a single list of data. Then, those data will be fed to a highly traditional biological neuron-inspired model, so-called fully connected neuron or dense layer, as shown in Fig. 6B. Moreover, multiplication and addition repeat until the model converges to the size of the desired class output. A simple LeNet-5 model (Liew et al., 2016) as depicted in Fig. 6C shows the end-to-end structure of a typical CNN, from the input image to output class.

Its cascaded arithmetic operation is where the CNN algorithm execution stressed the modern processor. It either spends too much processor time to serialise the process, or taking many hardware resources for parallelisation. The convolution arithmetic does multiplication and addition exhaustively. If the matrices of scanning windows are large or the network is deep/wide, then the computational demands required are high. As the multiplication and accumulation operations increase, memory access bottlenecking becomes the major limitation for DNNs (Capra et al., 2020). Traditional computing also uses floating-point units (FPU) which takes a wide area with high power consumption due to the colossal amount of logic gates involved. As the edge computing gains interest as the future trends of computation, energy efficiency has become a major concern for the CNN development and urged the researchers to rethink another way to process the information efficiently. Most of the modern FPU is of 32-bit floating-point (full precision). Thus, reducing the precision to 16 bits (half precision) or lower is one of the ways to improve CNN computation efficiency without much accuracy degradation.

BNN

In an extreme case, the parameters are reduced to only a single bit representation. This radical simplification of CNN is called BNN and gained attention among researchers in the industry due to its compactness in memory usage and practicality (Simons & Lee, 2019). In BNN, the parameters can only have two possible values, that is, −1 or 1. Despite some considerable degree of accuracy degradation, BNN does have several unique advantages. First is its model size; for instance, 64 MB of parameter data can be reduced to 2 MB, thus allowing the deployment of small embedded systems. Its little memory usage also allows memory-centric computing where the parameters can be stored directly beside the processing elements, thereby speeding up the computation by eliminating data communication cost (Angizi & Fan, 2017). Second is its hardware implementation capability. BNN requires some amount of arithmetic logic unit (ALU) to process fixed-point image data at the frontend (still cost less than FPU). However, the multiplication of the hidden layer can be simply an array of XNOR gates because the multiplication of −1 and 1 is of bipolar math. The high hardware utilisation of BNN in FPGA results in high throughput, whereas being an order of magnitude if not more energy-efficient than CPU and GPU despite lower clock speed (Nurvitadhi et al., 2017). Another unique advantage is that BNN is less susceptible to adversarial attack with stochastic weight quantisation in the hidden layer (Galloway, Taylor & Moussa, 2018). The adversarial attack is where data, for example, an image, are injected with noise and adversely affect the output class decision of a fine-tuned CNN model, albeit the doctored image has no perceptual difference to human eyes.

However, non-linear functions become useless due to the extreme information loss by the parameter quantisation. Instead, a threshold function can simply replace the normalisation and activation functions (Simons & Lee, 2019). The BNN also suffer accuracy degradation from highly challenging datasets, such as ImageNet classification, due to extreme information loss. As many studies explore for better BNN optimiser algorithms, Zhu, Dong & Su (2019) found that training optimisers might not help much due to BNN insensitivity to small changes in value. Instead, BNNs in parallel with ensemble technique (multiple trained BNNs in parallel and final decision with a majority vote) is a perfect fit, improving the overall BNN accuracy on large image classification datasets.

SC CNN vs BNN

The evolution of CNN to BNN challenged the idea of SC due to the competitiveness in hardware implementation capability. SC implementation is technically more challenging than BNN due to various custom logics and substantial uncertainty in future community support. After all, SC is still at its infancy in the CNN domain. Regardless of the different intentions and directions of development of SC and BNN, both studies try to explore alternatives for a highly efficient computing paradigm in the future of the IoT edge computing. With the massive exploitation and integration of DNN algorithms into small or remote devices, such as a smartwatch or surveillance camera, both fields of studies will contribute to the development of a highly realistic edge computing ecosystem.

SC implementation in CNNs

SC is considered the next frontier in energy-efficient edge computing (Jayakumar et al., 2016) due to its energy-efficient operation and ability to tolerate errors in domains of recognition, vision, data mining and so on. Meanwhile, many applications attempt to offload challenging workloads from cloud computing to edge devices. Thus, SC had become the hotspot of research interest.

Integral SC: a radical change in SC methodology for the sake of CNN

CNN is very popular in vision application due to its simplicity and accuracy. However, SC does not provide out-of-box experience as SC is not yet customised and explicitly optimised for the CNN algorithm. Hence, Ardakani et al. (2017) proposed a radical idea to use an integer stream instead of the traditional bitstream. The stochastic byte is ∈ [0,1,2…] so that to repurpose simple binary multiplier and bitwise AND as shown in Figs. 7B and 7C to process integer number in the stochastic domain, or integral SC. The idea is to preserve information across different precisions within a limited stochastic length.

Figure 7 Integral SC methodology.

(A) High precision stochastic number can be represented with shorter stream length with integer value. (B) Binary radix multiplier as integral SC scaled adder. (C) Modified MUX as integral SC multiplier. (D) Integer SC neuron block.

The effect of information loss becomes apparent when many MUX half-adding many stochastic streams exist together. The resultant precision requirements will only increase and require long overall bitstreams to preserve the precision of the half-added stochastic number. For example, a value of 0.5625 (9/16) requires a 16-bit length stochastic stream, whereas 0.875 (7/8) only requires 8-bit length. Although 0.875 can be expressed in 16-bit length, half-adding both numbers result in 0.71875 (23/32), or at least 32-bit length to preserve the output precision in the stochastic domain. If so, the overall stochastic bit length will need to be extended to 32-bit length. Cascading MUX adders in the CNN convolution stage will drive up the bit length requirements drastically, thus incurring considerable computational latency. The same problem also applies to the multiplier.

Then, the integral SC comes into play. Take Fig. 7A as an example. A value of 0.5625 can be effectively represented in the same length as the 8-bit length value of 0.875. Given that integral SC can preserve the stochastic information in an integer value, the final batch adding operation in CNN can be processed with tree adder as shown in Fig. 7D, eliminating parallel counter. Integer stream also allows short stochastic stream length, thus speeding up the SC time. They also proposed integer version of TanH k-state FSM because the traditional stochastic TanH (Stanh) function on FSM only accepts stochastic bits, thereby leading to the modern TanH FSM design. However, integral SC only solved the precision degradation issue, and many other CNN functions are yet to translate to SC domain. Moreover, the usage of binary adder and multiplier may not scale well in the case of deploying large CNN models. They claimed energy saving of 21% compared with the full binary radix computing but is still far from the expected power reduction in the SC transition.

Extended stochastic logic (ESL): another radical approach

ESL made an extreme modification to the SC methodology if integral SC is not radical enough. Instead of using a single stochastic bitstream for a value, ESL used two stochastic streams such that their ratio of division represents the actual value (Canals et al., 2016). ESL intends to compute the entire range of real numbers in the stochastic domain. For example, if x* is a whole number, then x* = p*/q*, where p* and q* are the ESL stochastic pair for x*. p* and q* remain in real number ∈ [−1,1] in the bipolar format, but obtaining its ratio x* can translate to the entire range of real numbers ∈ [− ∝,  ∝].

ESL requires dedicated logic gate for p* and q* stochastic streams. Taking Figs. 8A and 8B as an example, if x* = p*/q* and y* = r*/s*, then by probability math, multiplication between two separable stochastic streams will be: (8) x∗×y∗=p∗×r∗q∗×s∗=t∗u∗,

where t* and u* are the output pair of stochastic streams. Division can be done simply by flipping the nominator and denominator of the second stochastic pair. In the case of stochastic addition, the stochastic pair can be processed such that: (9) x∗+y∗=p∗q∗+r∗s∗=p∗×s∗+q∗×r∗q∗×s∗=t∗u∗,

whereas subtraction can be done by NOT gate inversion as shown in Fig. 8C.

Figure 8 ESL arithmetic unit.

(A) ESL multiplier. (B) ESL divider by crossing multiplication. (B) ESL adder and subtractor circuit.

Value splitting is feasible in the stochastic domain due to the nature of probabilistic computing. However, splitting into double stochastic streams complicated everything, including a compulsory custom bipolar divider (convert t* and u* back to real number representation) before bipolar TanH function blocks. The custom block extensively used comparator and RNG, which add a red flag for efficient computing. The neural network may compute in the real number ∈ [− ∝,  ∝] on the early day, but the CNN nowadays commonly compute in bipolar math. After all, the final output class of CNN only need to tell the computer whether the probability ∈ [0,1]. ESL did provide an insight into how SC can perform normal arithmetic full-range computation. However, verifying whether ESL is better than other SC methods for CNN use case despite the attractive circuit simplicity in primary arithmetic operations is hard due to the non-linear activation function complexity in ESL implementation.

Approximate parallel counter (APC) and Btanh: a simple and energy-efficient approach

Implementing radical changes in every SC component might not be easy. Thus, another highly effective approach with traditional stochastic bitstream is APC. Other than the frontend binary to stochastic conversion stage of SNGs, the final stochastic to binary conversion stage is also equally important (Kim, Lee & Choi, 2016a; Kim, Lee & Choi, 2016b).

In the case of accumulating multiple bitstreams, MUX adder could become inaccurate due to loss of n −1 bits input information (Li et al., 2017c). In this case, a parallel counter as the one in Fig. 9B is preferred consisting of an array of full adders (FA), but FA is relatively expensive as it uses binary adder logic circuits. The accurate parallel counter should no longer be used as SC is already based on approximate computation. Thus, an APC has been proposed to reduce the FA components with a slight trade-off on accuracy whilst achieving the same counting function at lower area and power consumption as shown in Fig. 9A. The proposed APC could save area and power by 38.3% and 49.4%, respectively. The caveat is that the output from APC is in the binary domain; thus, directly removing any stochastic stream from the stochastic domain computation.

Figure 9 SC bitstream accumulation.

(A) APC. (B) Accurate parallel counter. (C) Accumulation and Btanh activation workflow.

Although the traditional Stanh uses single input k-state FSM, with the inspiration from integral SC research, the binary output from APC is cleverly reused as an input for another modified binary input FSM called Btanh. TanH activation function is essential in CNN. For example, if the binary output value is 4, then the FSM will directly jump four states instead of step-wise jumps in Stanh. More granular Tanh step-function could also be achieved, which is not possible with Stanh FSM. In the end, the binary output values from APC will be indirectly converted back to stochastic stream with TanH non-linear function applied, completing the stochastic convolution computation as depicted in Fig. 9C. Moreover, energy usage can be further reduced by 69% by sacrificing 1.53% of accuracy with EDT, that is, terminating computation at 50% of the computing time. Then, their APC and Btanh components had become the foundation for other SC CNN approaches in the next coming years.

Near-perfect SC implementation in CNN algorithm

Ren et al. (2016), Li et al. (2017c); Li et al. (2017b), Ren et al. (2017) and Li et al. (2018a) proposed a complete overview of a near-perfect CNN analogy in the SC domain, including the following: the missing pooling layer, ReLU and sigmoid activation layer, and normalisation layer which will be discussed separately in the sub-sections below.

SC average pooling and max-pooling layers

The purpose of CNN pooling layer is to reduce memory usage and reduce model size. Ren et al. (2016) first used cascaded MUX as the average pooling function in CNN as shown in Fig. 10A. This solution is simple but may face the precision loss issue as described in the Integral SC research. Average pooling may not help in CNN training convergence either. Ren et al. (2017) proposed max-pooling hardware equivalent to the widely adopted CNN max-pooling layer. The stochastic stream with a maximum value at any given time in the stochastic domain could not be verified. Hence, a dedicated counter for each stochastic stream is required to sample and evaluate which stream is of maximum value. By referring to Fig. 10B, the counter samples the first few bits and compare the magnitude at the end of bitstream sampling to make an early decision on which stochastic stream should be chosen to continue the path. The first few bit information could be inaccurate and thus is an approximate max pooling. Nevertheless, the decision will eventually converge to the bitstream of maximum value if the sampling continues due to the properties of SC progressive precision. Moreover, if the bitstream is long, then less information will be lost, thereby achieving negligible accuracy loss.

Figure 10 SC pooling function.

(A) 2 × 2 average pooling with cascaded MUX adder. (B) hardware-oriented approximate max pooling circuit. (C) Stochastic MAX function, cascading them will create pure SC max pool block.

However, a more straightforward stochastic max-pooling block was proposed by Yu et al. (2017). With only an XOR gate, FSM and MUX, a novel stochastic MAX block could select whichever stream of higher value. With XOR gate controlling the FSM state jumping, the probability of the opposite stream could be inferred from another bitstream by generating the condition of bit entanglement. As such, whenever the FSM sampled a 0’s bit from the current bitstream, it implies a 1’s bit on the opposite bitstream. Thus, whenever inequality between two bitstreams exists, the FSM state will be biased to the one with higher magnitude, completing the MAX function with the MUX. Cascading the MAX function block could realise the max-pooling function block as shown in Fig. 10C.

SC ReLU and sigmoid activation layer

The CNN activation layer is similar to the usual neuron activation function. ReLU function, as the name suggests, performs rectification and cutting off any negative value such that: (10) fx= max0,x.

ReLU function is trendy due to its fast computation and solves diminishing return in backward propagation learning during the CNN training stage. However, no SC equivalent circuit existed for that particular function; thus, Li et al. (2018a) proposed a novel SC-based ReLU function block as depicted in Fig. 11A.

Figure 11 Other SC activation functions.

(A) ReLU activation function. (B) SC sigmoid activation function with bias input.

Firstly, the ReLU amplitude will be naturally maxed out at value = 1 in the stochastic domain, but this is not a concern as clipped ReLU has no significant accuracy degradation (Fei-Fei, Deng & Li, 2010). Secondly, a negative value must be clipped to zero. Notably, the number of 0’s bit in the bipolar stochastic stream determines the magnitude of negativity. Thus, when the accumulated value is less than the reference half value (the 0’s bit is more than 1’s bit) in a given sample time, the output will be forced to be 1’s bit. Otherwise, the output will follow the pattern of emulated linear function from the FSM. Although real number convergence in the accumulator takes time, the real value information is equally distributed in the stochastic bitstream. Hence, obtaining an accurate comparison is possible by observing the first few bits of information; thus, inaccuracy is negligible. Moreover, the comparison is synchronous to the input; therefore, no latency will be incurred.

In the case of larger and deeper CNN models such as VGGNet and GoogleNet, the sigmoid function becomes more favourable as non-linear function. As such, Li et al. (2017a) proposed a hardware-oriented SC sigmoid approximation function as shown in Fig. 11B. Since the output of the stochastic stream is maxed at 1, the Taylor series expanded sigmoid function could be approximated as: (11) 11+ exp−x≈1,x>212+14x,−2≤x≤20,x<−2.

By strategically partitioning the positive summation and negative summation in such a way that: (12) A=14∗∑posP⋅Q+12+bias+4,B=14∗∑negP⋅Q+bias−4,

the approximate stochastic sigmoid activation function could then be realised by subtracting both parts such that: (13) A−B=12+14∑P⋅Q+bias,

where ‘P’ and ‘Q’ are the weight and pixel value respectively. Therefore, by pre-scaling the weights and bias to quarter times, the stochastic sigmoid function could be devised as a result, with the added benefit of including bias information which is missing in the previous SC CNN implementation. The binary adder now is the sigmoid activation function itself, eliminating the need for extra hardware cost such as FSM. However, unlike the APC + Btanh function block, the accurate parallel counter is needed.

The sigmoid function is not limited to CNN algorithm, or rather, is a universal activation function in other DNN classifier algorithms such as multilayer perceptron and restricted Boltzmann machine. With 1024-bit length stochastic stream, the proposed SC sigmoid activated convolution neuron block could perform as accurate as binary computing CNN while consuming 96.8% and 96.7% less area and power respectively, hugely improving the capability of SC in the DNN algorithm computation in general.

SC normalisation layer

The purpose of the normalisation layer is to reduce internal covariance, thereby improving the overall CNN output accuracy. If the ReLU activation is applied to the previous layer, only a simple local response normalisation function is required, which can be summarised as: (14) bx,yi=ax,yik+α∑j=max0,i−n∕2minN−1,i+n∕2ax,yj2β,

where the summation part accumulates all N numbers of adjacent neuron output of ax,yi. ‘k’, ‘n’, ‘ α’, and ‘β’ are hyperparameters which can be determined by CNN backpropagation training. The complexity of the mathematical relationship can be decoupled into three compute components, square and sum (calculate the denominator components), exponential function with “β” and finally division. Li et al. (2017c) used stochastic square, FSM activation block and traditional gradient descent SC divider to construct SC normalisation circuit as shown in Fig. 12 to perform SC normalisation. The accuracy had improved with SC normalisation function and only dropped by 0.88% compared with the original binary AlexNet CNN model, achieving six times in the area and five times in power savings compared with binary equivalent normalisation. However, they could have utilised newer SC divider as discussed in the basic concept section.

Figure 12 Normalisation unit in SC CNN.

Figure 13 Binary interlaced SC, where SC is used as MAC accelerator.

(A) SC MAC unit block. (B) SC MAC optimisation by cutting off SC early with advancing weight bits. (C) Novel SNG with MUX and FSM.

Other optimisations

The dropout layer is one of the regularisers in CNN to prevent overfitting. However, dropout layer functions only at the CNN training phase, and no custom hardware adaptation is needed at the inference stage, hence no extra hardware overhead. Li et al. (2018a) optimised the APC function block by utilising inverse mirror FA to reduce the number of transistors required for single FA from 32 to 24 transistors. They also proposed the APC design which input is not a power of two by incorporating inverse half adder. APC optimisation further reduced the area required by at least 50% and an average of 10% improvement in energy efficiency.

In terms of SC accuracy, the bipolar format remains the major limitation as bipolar is generally worse than the unipolar in terms of SC accuracy (Ren et al., 2016). To overcome the signed value accuracy limitation, Zhakatayev et al. (2018) decoupled the sign information from the stochastic stream and added one stochastic bit pair specifically to store the sign value. Unlike stochastic probability value, the sign value of a stochastic stream is deterministic, thus, can be processed separately from the stochastic magnitude. Although small hardware overhead is needed to process the sign function, such as an extra XOR gate to multiply signed value, the accuracy gain is significant, 4∼9.5 times better compared to the bipolar format. With that advantage in mind, the little extra hardware cost for sign processing is trivial.

Binary Interlaced SC, two is better than one

Full-fledged SC CNN might not be feasible to fit a wide variety of modern complex CNN models. However, the massive multiplication parallelism of SC is still very favourable. Thus SC-based multiply-accumulate (MAC) unit was proposed by Sim & Lee (2017) as shown in Fig. 13A to act as multiplier accelerator for binary computing. The MAC leverages the parallelism of SC multiplier, then accumulate value with accurate parallel counter, returning pure binary value to other binary computing circuits at the end of SC cycle. This approach, while not the most energy-efficient one, achieved two times the area efficiency and at very high throughput compared to binary computing. With only a single layer SC in mind, Sim et al. (2017) further leveraged the SC MAC to perform unipolar SC multiplication. All the stochastic 1’s bit of the neuron weight value was pushed ahead of time by down counting the weight value so that the SC cycle could terminate when the stream tail of the weight ended with 0’s bit as depicted in Fig. 13B. This event is possible because any section of the stream could represent the true value of the stream due to the probabilistic nature. It is technically feasible as long as single layer SC is concerned. They also proposed a novel MUX FSM based SNG. By predefining the MUX selection sequence in such a way that the output is the sum of binary weight, the binary input could be directly converted into a stochastic stream as depicted in Fig. 13C, eliminating the need of WBGs which could be expensive in FPGA implementation. With the strategic down-counting timing, an area-delay product reduction of 29%∼49% is achieved while being 10%∼29% more energy efficient compare to binary computing. In any case, they ignored the SNGs hardware overhead in performance comparison.

Considering that only a single SC layer is required, Hojabr et al. (2019) radically redesigned the MAC unit by exploiting computing pattern in modern CNN design and proposed Differential MAC (or DMAC). Firstly, because CNN ReLU function always returns positive value, in addition to the binary pixel of positive value, thus, up/down counter could be used as ReLU function. Secondly, considering that a pixel value will eventually pass through all the weight multiplication matrix of CNN scanning window in the convolution process, the neuron weights could be sorted offline ahead of time. In this way, the weight differential from the next sorted weight of higher value is guaranteed to be positive, thus, can be fed to a down counter similar to SC MAC to pipeline the stochastic multiplication. Since the first weight is of minimum value which could be negative, a D Flip-Flop is used to hold the sign information just for the first bipolar multiplication. Thus, multiplying in SC is as simple as counting the number of bits from the MUX AND-ing with counter ‘enable’ control from the weights as depicted in Fig. 14. The FSM could be shared among all MUX, ignoring the stochastic correlation issue because the multiplication is mutually independent (Yang et al., 2018). The buffered accumulated value will then continue the summation operation as the DMAC final stage. This major circuit overhauling could deliver 1.2 times and 2.7 times gains in speed and energy efficiency respectively relative to the former MAC with the benchmarking on more modern CNN models.

Figure 14 Differential MAC. Major overhauling to the SN MAC with counter and differential weight control indexing to pipeline the SC MAC computation.

Stochastic quantisation, SC is going asynchronous

In the face of quantised binary CNN whereby the arithmetic is lower than 8-bit precision, no optimisation had been done on the SC CNN counterpart. SC could consume a lot of logic gates as well, especially in CNN use case. Thus, Li et al. (2018b) proposed a novel multiplier with shifted unary code (SUC) adder. From the binary interlaced SC research, the weights do not have to follow probability distribution as the pixel value does, as long as the next SC component is not computing in the stochastic domain. By strategically using the weight information as a timing control for SC multiplication, meaningful bits from each stream could be quantised and unified into a single multiply-sum-averaged stochastic stream by OR-ing the parallel bitstreams asynchronously as depicted in Fig. 15. The SUC adder significantly reduced the requirement of parallel counter whereby its internal FA is expensive in the perspective of SC. The area and power savings are significant as a result, as much as 45.7% and 77.9% respectively relative to usual unipolar SC with less than 1% accuracy loss compared to quantised binary CNN, paving the way for more efficient parallel counting accumulation mechanism in SC CNN.

Figure 15 Stochastic quantisation accumulation with SUC adder.

(A) The different information portion of the stochastic stream could be encoded into a single stream by OR-ing the required bitstream asynchronously. (B) SUC paired with SC sigmoid activation function.

Analog-to-Stochastic Converter, SC CNN is ready to be embedded

In the case of direct interfacing with analogue input, such as analogue camera sensor, Analog-to-Digital Converter (ADC) is usually being deployed, but at the cost of requiring memory storage. Zhang et al. (2019) proposed a novel converter, namely, Analog-to-Stochastic converter (ASC) as shown in Fig. 16A where the analogue voltage differential could be directly decoded into stochastic streams with thermometer encoding scheme. The stochastic stream could either be encoded via LFSR, counter, or newly proposed thermometer coding as depicted in Fig. 16B. The thermometer coding is capable of generating parallel bit streams at once but has higher error compared to the others. Nevertheless, with long enough bitstream length, those error is negligible. The thermometer encoding enabled the design of novel ASC which allows SC CNN to be directly interfaced with analogue voltage input, eliminating ADC and memory storage.

Figure 16 ASC with thermometer coding.

(A) Implementation of ASC on thermometer-encoded SC circuit, eliminating the need for ADC and memory components. (B) Thermometer coding could be utilised for SNGs.

SC CNN is meant for memory-centric computing

Notably, SC CNN does require a tremendous amount of weight data similar to fixed point binary CNN. Despite many SC CNN architecture innovations, however, without efficient weight storage near to SC elements, SC CNN will suffer memory bandwidth bottlenecking similar to the binary computing. Since the weight information is fixed from the training process, those data can be stored in a more area and power-efficient non-volatile Domain-Wall Memory (DWM) (Ma et al., 2018) built beside the SC elements. This strategy could eliminate memory bandwidth bottlenecking by bringing memory closer to the computing element, namely, memory-centric computing or in-memory computing. SC CNN can greatly benefit from memory-centric architecture due to the nature of massive parallelism. Memoisation approach could also be executed in memory-centric design by storing the weight data directly in a predefined stochastic bitstream representation instead of original binary values. As such, sequential read of stochastic bit from DWM could use less energy while reducing the SNGs usage. Further area reduction could be achieved by sharing APC and weights. Thus, an area and power reduction of 52.6% and 17.35 times were reported respectively relative to standard SC CNN as a result of resource sharing and more efficient memory-centric architecture in the SC CNN circuit.

SC implementation in BNN: the best of both worlds

As mentioned earlier in the basic concept section, BNN challenged the existence of SC circuits in CNN computing. As the saying goes, the enemy of an enemy is a friend, and considering that SC and BNN target efficient CNN computation, why not combine both to maximise the benefits from both aspects, which is what (Hirtzlin et al., 2019) precisely targeted for. The inspiration for this particular approach is that the SC and BNN come into the same conclusion that XNOR gate can be used as a bipolar multiplier, as depicted in Fig. 17A, despite different directions of development. If somehow a way to process the BNN model in stochastic mean exists, then the SC can take a free ride to the BNN’s internal logic.

Figure 17 SC BNN methodology.

(A) The similarity of SC and BNN in terms of logic gate utilisation. (B) Usual configuration in binary BNN. (C) SC BNN first layer binary image conversion in SC BNN.

Although BNN process information at the bitwise level in the hidden layer, the initial layer still needs to deal with input images of fixed-point binary number as shown in Fig. 17B. In most cases, ALU is utilised for real number calculation, or digital signal processing unit in the case of FPGA. They attempted to fuse the SC domain onto the first layer by translating image input into stochastic bitstreams and then exploiting SC logic similarity in BNN for bipolar multiplication to take advantage of the BNN logic. However, unique data pre-processing is needed so that the trained network is trained on a serialised stochastic binary image instead of the original grayscale image. The input image is converted into multiple stochastic image representations as shown in Fig. 17C where the bitstream generation of each pixel follows the function of SNG. Then, the number of stochastic images generated is equal to the stochastic bit length of the data. A ‘popcount’ accumulator is implemented at the end of the layer to restore the real number before proceeding to the next threshold function, which had replaced the activation function and batch normalisation. The difference of their BNN usage compared with the general BNN is that they treated the BNN XNOR gate as if it is of SC CNN stochastic logic. Notably, the SC only apply on the first layer, and the rest of the hidden layer still follows BNN logics.

In the end, they claimed to have 62% area reduction whilst only suffer 1.4% accuracy degradation in Fashion-MNIST dataset classification compared with the binary first-layer BNN. They also claimed that with three stochastic image representations, SC BNN could achieve the same performance as binary BNN implementation at 2.4 times lower energy usage, which is very similar to the EDT approach. They even extended the experiment with advanced CIFAR-10 images with RGB channels. By following the same image conversion principle in channel-wise, the SC BNN achieved the same accuracy as full binary BNN, proving that eliminating ALU at the first BNN layer is possible. Nevertheless, one possible confusion is that they could have mistaken the BNN weight information as part of the stochastic domain. The BNN weights were trained in the binary domain with images of real fixed-point value, but it is not a concern as long as the BNN weights are represented in fully quantised ‘−1’ or ‘1’ vector regardless of the computing domain.

Discussion

We discussed the SC CNN and BNN elements in component-wise. However, a visualisation approach is necessary to obtain the full picture of how are they exactly being stacked together as SC CNN and SC BNN, which no one had emphasised on in almost all related studies. Otherwise, novel readers might be having a hard time to grasp the idea and motives behind the effort of SC development, particularly for those studies mentioned above with the mixed bag of vastly different fields of study.

SC CNN and SC BNN from a holistic perspective

Modern computing handles the CNN computation by aggregating all values layer-by-layer until the final class output is converged. The hidden truth behind the oversimplified drawing of CNN as in Fig. 6C is that there could have a lot of data accumulation and transfer between the processor and memory. Even if modern GPUs could parallelise thousands of arithmetic operations, it still takes time to buffer computed data into local memory for each feature map or layer, because it is impossible to read and write on the same memory at the same time.

Conversely, SC handles the computation information in a different approach as depicted in Fig. 18. Due to the extreme parallelisation capability of the SC circuit, all of the data could be technically preloaded into local memory before the starting of the SC cycle. Although stochastic stream could take hundreds or even thousands of clock cycles to complete (each clock for each stochastic bit), SC pipelined all CNN arithmetic operation from top-down. Thus, all of the bits at a particular moment passed though all CNN layers at every SC clock cycle. If a clock cycle took 1 µs, then a full-fledged SC CNN inference with 1-kilobit length stochastic streams could, in theory, complete the CNN computation in under 1 ms. By then, a new full-sized image data could have been buffered asynchronously readily available for the next SC cycle. Thus, in the perspective of the SC circuit, memory bandwidth bottlenecking might not be an issue. The simple computing elements in SC allow large-scale parallelisation, which is incredibly favourable to CNN hardware implementation in edge computing application. The advantage will only be highly prevalent when noise tolerance is essential at a higher clock speed in the future of computing or deployment of a big CNN model which requires larger data parallelisation.

Figure 18 Process flow in SC CNN and the internal computing domain interchange.

In the case of SC BNN as illustrated in Fig. 19, the converted stochastic images could exploit the BNN XNOR logic for SC, eliminating the need for ALU. Although the SC domain ended at the first layer, the subsequent BNN bipolar multiplication, accumulation and threshold loops do not take much computing time either, virtually single-layer pass in one or few clock cycles. Given the nature of the layer-wise operation, BNN could in practice allow layer folding, that is, reusing the computer components of the previous layer by reloading weight information (Mittal, 2020), further reducing the area and power required which are not possible on SC CNN. SC BNN also allows in-memory computation because those bit weights can be stored right next to the computing gate arrays, further improving energy efficiency by eliminating the cost of communication bandwidth. The ensemble technique on BNN could also perform as accurate as full precision DNN (Zhu, Dong & Su, 2019). Thus, the area and power savings of SC BNN could be extreme, challenging the performance of SC CNN.

Figure 19 Process flow in SC BNN, stochastic image generation methodology and the internal computing domain interchange.

Table 1 Performance difference across SC and conventional binary domain.

CNN Model	Platform	Year	Method	Area (mm2)	Power (W) or energy (nJ)	Accuracy (%)	Energy efficiency (images/J) or (GOPS/W)	
Lenet-5	CPU	2009	Software	263	156 W	99.17	4.2	
	GPU	2011	Software	520	202.5 W	99.17	3.2	
	ASIC	2016	SC 256 bit (Ren et al., 2017)	36.4	3.53 W	98.26	221,287	
	ASIC	2018	SC 128bit (Li et al., 2018a)	22.9	2.6 W	99.07	1,231,971	
ASIC	2018	SC DWM 128bit (Ma et al., 2018)	19.8	0.028W	98.94	–	
AlexNet (last second layer)	CPU	2009	Software	263	156 W	–	0.9	
	GPU	2011	Software	520	202.5 W	–	2.8	
ASIC	2018	SC 128bit (Li et al., 2018a)	24.7	1.9 W	–	1,326,400	
Custom (3x3filter)	ASIC	2015	Binary	5.429	3.287mW	–	–	
	ASIC	2017	SC MAC	1.408	1.369mW	–	–	
ASIC	2019	SC DMAC	1.439	1.393mW	–	–	
Custom (Ardakani et al., 2017)	ASIC	2017	Binary	–	380 nJ	97.7	–	
ASIC	2017	Integral SC	–	299 nJ	97.73	–	
ConvNet for MNIST	ASIC	2015	Binary	0.98	0.236W	–	1158.11 GOPS/W	
ASIC	2017	SC MAC	0.43	0.279W	–	5640.23 GOPS/W	
Custom (Hirtzlin et al., 2019)	ASIC	2019	BNN	1.95	220 nJ	91	–	
ASIC	2019	SC BNN	0.73	90 nJ	89.6	–	
Notes.

GOPS Giga operations per second

Table 2 Component-wise performance comparison of SC CNN.

SC CNN/BNN components	Author	Platform/ software	Relative accuracy (%)	Area/gate count (%)	Power/Energy saving (%)	Relative to	
Integra SC	Ardakani et al. (2017)	FPGA & ASIC	+0.03	−33.9	21.3	Binary computing	
ESL	Canals et al. (2016)	FPGA	−2.23	–	–	Binary computing	
APC + Btanh	Kim, Lee & Choi (2016a), Kim, Lee & Choi (2016b) and Kim et al. (2016)	Synopsys Design Compiler	−0.18; −1.71 (EDT)	−50.0	70.0; 76.2 (EDT)	Binary computing	
APC with inverse adder	Li et al. (2018a)	Synopsys Design Compiler	–	−50.0	10.0	Normal APC	
SC MaxPooling	Ren et al. (2017)	Synopsys Design Compiler	−0.20	−92.7	98.3	GPU computing	
SC ReLU activation	Li et al. (2018a)	Synopsys Design Compiler	−0.88	−95.3	99.1	GPU computing	
SC normalisation	Li et al. (2017b)	Synopsys Design Compiler	−0.02	−83.8	88.9	Binary computing	
SC MAC	Sim & Lee (2017)	Synopsys Design Compiler	−1	−93.9	−89.4	Binary computing ASICa	
SC DMAC	Hojabr et al. (2019)	Synopsys Design Compiler	–	−73.5	292	AlexNet	
					147	InceptionV3	
					370	VGG16	
12	MobileNet	
SC Sigmoid activation	Li et al. (2017a)	FreePDK	−0.01	−96.8	96.7	Binary computing	
SC Quantization	Li et al. (2018b)	FreePDK	−0.79	−98.6	99.1	Binary computing	
			–	−45.7	77.9	Unipolar SC	
–	−60.3	85.8	Bipolar SC	
SC BNN	Hirtzlin et al. (2019)	Cadence First Encounter	−1.40	−62.0	240	Binary BNN	
Notes.

a Binary computing ASIC apply to the CNN model comparison.

Although no standard reference exists for a fair comparison, we can compare the performance difference of SC CNN/BNN in CNN model-wise as shown in Table 1 to highlight the clear advantage of SC in CNN application. Nevertheless, the year of comparable studies varies greatly, and hardware and software efficiencies had greatly improved over the last decade, thus should only be taken as a rough comparison. In the case of component-wise performance comparison, Table 2 could further clarify the performance number that had been mentioned in the previous section if any.

Conclusions

The SC may still not well developed relatively speaking. Still, with the trending of highly parallelised computing use case, SC might be the good old yet not-so-old idea, specifically when people are still actively researching and optimising SC circuits with the driving momentum of CNN algorithm. That being said, the FPGA itself is still not widely adopted in the programming community, let alone the SC adaptation. Numerous efforts were made in the high-level CNN to FPGA translation for binary domain computation (Liu et al., 2017; Noronha, Salehpour & Wilton, 2019). However, the bridging effort of SC in FPGA is near to non-existence or should be said most of the SC studies lean to ASIC. Many people are interested in offloading computationally intensive workloads such as image processing and CNN inferencing to the co-processor. Thus, SC elements should be made an open-source IPs and introduced into the FPGA design ecosystem so that people can innovate on it. The open-sourcing design could help accelerate the SC development because researchers do not have to redesign the IP from scratch which is the major hurdle for novel development and could turn down people from being interested in SC technology. It could be the primary reason why SC CNN lacks attention, leading to a low number of comparable data as well as benchmarking.

Speaking of parallelism capability in SC, data bandwidth bottlenecking could be a major challenge. Even though SC can have vast arrays of WBG or comparator to compare a massive amount of binary values at once, delivering massive data on time is challenging. Notably, SC does require hundreds if not thousands of clock cycles to complete. Thus, data transfer could be pipelined and buffered asynchronously. Moreover, a tremendous amount of data needs to be ready beside the SC elements. As such, local memory element such as SRAM (in ASIC terms) or BRAM/Flip-flop (in FPGA term) limitation should be the concern. In any case, memory-centric computing design should be the direction of SC development, especially in SC CNN, where hundreds of thousands, even millions of operations could be parallelised.

There are still a lot of optimisation rooms for SC implementation on FPGA since most of the modern FPGA consists of 6-input lookup tables. ASIC logic might not be able to translate into the FPGA fabric efficiently because lookup tables are hardwired. Although FPGA is flexible in terms of hardware implementation, it is not as customisable as the ASIC. Modern FPGA also consists of other resources capable of performance computing such as digital signal processors or arithmetic logic awaiting to be utilised. However, those aspects could only be discovered in future research efforts.

Nomenclature

ADC Analog-to-digital converter

ALU arithmetic logic unit

APC approximate parallel counter

ASC Analog-to-stochastic converter

ASIC application-specific integrated circuit

BNN binarised neural network

Btanh binary input Stanh

CNN convolutional neural network

CPU central processing unit

DNN deep neural network

DWM domain-wall memory

EDT early decision termination

FA full adder

FPGA field-programmable gate array

FPU floating-point unit

FSM finite state machine

GPU graphic processing unit

IoT internet of things

LFSR linear feedback shift register

MAC multiplier-accumulator

ReLU rectified linear unit

RNG random number generator

SC stochastic computing

SNG stochastic number generator

SUC shifted unary code

Stanh stochastic TanH

TanH hyperbolic tangent

WBG weighted binary generator

Additional Information and Declarations

Competing Interests

Author Contributions

Data Availability

The authors declare there are no competing interests.

Yang Yang Lee conceived and designed the experiments, performed the experiments, analyzed the data, prepared figures and/or tables, authored or reviewed drafts of the paper, and approved the final draft.

Zaini Abdul Halim analyzed the data, authored or reviewed drafts of the paper, and approved the final draft.

The following information was supplied regarding data availability:

No raw data is available for literature review.

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
