# Peer review of "Stochastic computing in convolutional neural network implementation: a review"

_PeerJ Computer Science, doi:10.7717/peerj-cs.309_

## Round 0.1 · original submission · Major Revisions

Please revise the paper according to the comments, Thank you.

Reviewer 1 ·

Basic reporting

The article must be written in English and must use clear, unambiguous, technically correct text. The article must conform to professional standards of courtesy and expression.

Experimental design

The investigation must have been conducted rigorously and to a high technical standard. The research must have been conducted in conformity with the prevailing ethical standards in the field.

Validity of the findings

The conclusions should be appropriately stated, should be connected to the original question investigated, and should be limited to those supported by the results. In particular, claims of a causative relationship should be supported by a well-controlled experimental intervention. Correlation is not causation.

Additional comments

This paper is totally well written easy to understand because the authors did a good conclusion on the part of basic concepts. This manuscript mainly researches several questions such as developments of SC CNN and how does it compute etc. Thus, authors must be concerned about some details as follows:
1. I suggest the authors attach all the figures and place them around the appropriate paragraph respectively in the manuscript as much as possible. Anyway, authors must improve the quality for all the figures, font size must be adjusted and unified, the resolution must be improved. If all the figures are quoting, authors must indicate references respectively.
2. This manuscript has too many abbreviations and parameters. Therefore, if necessary, I suggest authors should list a nomenclature table with its name, unit of measurement, and description of all variables (explanation) at the beginning of this manuscript.
3. In the Basic Concepts section, some algorithms and methods such as logical formulas should be more concise. Some expressions should be modified to the mathematical mode when using exponential computing, for instance, 2n, instead of 2^n.
4. Related work is not enough in this paper, authors should quote the latest references (2019-2020). In addition, this manuscript is weak in part of the conclusion. It will be better if authors make a summary and extension for each research work in "SC implementation in CNN's".
5. What are the development prospects of the SC CNN algorithm? And how did other relevant researchers prepare to process the open problems? By the way, the author's personal views throughout this paper are quite weak, this section must be restructured and polished.
6. Because of the development of CNN architectures, various models based on CNN such as ImageNet and googLeNet etc. Authors need to compare the performance variation between state-of-the-art CNN architectures and traditional CNN during SC Implementation.

Reviewer 2 ·

Basic reporting

This submission reviews the key concepts of Stochastic computing (SC) and the circuit structure and then compare the advantages and disadvantages amongst different methods. Finally, This submission concludes the overview of SC in CNN and make suggestions
for widespread implementation in the near future.

Experimental design

This submission is within the scope of the journal

Validity of the findings

This is a review without technical novelty.

Additional comments

This submission is with good discussions.

Reviewer 3 ·

Basic reporting

English writing should be improved in this manuscript.

Experimental design

no comment.

Validity of the findings

no comment.

Additional comments

This review article introduces stochastic computing and deep learning, especially CNN based SC methods. This is quite helpful for these researchers in this field.

1, English writing should be improved. Some statistical results should be shown in tables instead of sentences.
2, More recent researches should be introduced for better description.

---

## Round 0.2 · accepted · Accept

The paper can be accepted as it is.

Reviewer 3 ·

Basic reporting

No comment.

Experimental design

No comment.

Validity of the findings

No comment.